# Reduced mortality in COVID-19 patients treated with colchicine: Results from a retrospective, observational study

**Lucio Manenti**[1]☯, **Umberto Maggiore**[1]☯*, **Enrico Fiaccadori**[1], **Tiziana Meschi**[2], **Anna Degli Antoni**[3], **Antonio Nouvenne**[2], **Andrea Ticinesi**[2], **Nicoletta Cerundolo**[2], **Beatrice Prati**[2], **Marco Delsante**[1], **Ilaria Gandoflini**[1], **Lorenzo Donghi**[3], **Micaela Gentile**[1], **Maria Teresa Farina**[1], **Vincenzo Oliva**[1], **Cristina Zambrano**[1], **Giuseppe Regolisti**[4], **Alessandra Palmisano**[1], **Caterina Caminiti**[5], **Enrico Cocchi**[6], **Carlo Ferrari**[3], **Leonardo V. Riella**[7], **Paolo Cravedi**[8], **Licia Peruzzi**[6]

**1** Dipartimento di Medicina e Chirurgia, Università di Parma e UO Nefrologia, Azienda Ospedaliero-Universitaria di Parma, Parma, Italy, **2** Dipartimento Geriatrico-Riabilitativo, Azienda Ospedaliero-Universitaria di Parma, Parma, Italy, **3** UO Malattie Infettive ed Epatologia, Azienda Ospedaliero-Universitaria di Parma, Parma, Italy, **4** Dipartimento di Medicina e Chirurgia, Università di Parma e UO Clinica e Immunologia Medica, Azienda Ospedaliero-Universitaria di Parma, Parma, Italy, **5** UO Ricerca e Innovazione, Azienda Ospedaliera-Universitaria di Parma, Parma, Italy, **6** Pediatric Nephrology Unit, Regina Margherita Children's Hospital, Città della Salute e della Scienza di Torino, Turin, Italy, **7** Division of Nephrology, Massachusetts General Hospital, Harvard Medical School, Boston, MA, United States of America, **8** Renal Division, Department of Medicine, Icahn School of Medicine at Mount Sinai, New York, NY, United States of America

☯ These authors contributed equally to this work.
* umberto.maggiore@unipr.it

**Data Availability Statement:** All relevant data are within the paper and its Supporting Information files.

## Abstract

### Objectives

Effective treatments for coronavirus disease 2019 (COVID-19) are urgently needed. We hypothesized that colchicine, by counteracting proinflammatory pathways implicated in the uncontrolled inflammatory response of COVID-19 patients, reduces pulmonary complications, and improves survival.

### Methods

This retrospective study included 71 consecutive COVID-19 patients (hospitalized with pneumonia on CT scan or outpatients) who received colchicine and compared with 70 control patients who did not receive colchicine in two serial time periods at the same institution. We used inverse probability of treatment propensity-score weighting to examine differences in mortality, clinical improvement (using a 7-point ordinary scale), and inflammatory markers between the two groups.

### Results

Amongst the 141 COVID-19 patients (118 [83.7%] hospitalized), 70 (50%) received colchicine. The 21-day crude cumulative mortality was 7.5% in the colchicine group and 28.5% in the control group (P = 0.006; adjusted hazard ratio: 0.24 [95%CI: 0.09 to 0.67]); 21-day

**Funding:** PC is supported by NIH NIAID grant 3U01AI063594-17S1. No external funding was received for this study.

**Competing interests:** The authors have declared that no competing interests exist.

clinical improvement occurred in 40.0% of the patients on colchicine and in 26.6% of control patients (adjusted relative improvement rate: 1.80 [95%CI: 1.00 to 3.22]). The strong association between the use of colchicine and reduced mortality was further supported by the diverging linear trends of percent daily change in lymphocyte count (P = 0.018), neutrophil-to-lymphocyte ratio (P = 0.003), and in C-reactive protein levels (P = 0.009). Colchicine was stopped because of transient side effects (diarrhea or skin rashes) in 7% of patients.

## Conclusion

In this retrospective cohort study colchicine was associated with reduced mortality and accelerated recovery in COVID-19 patients. This support the rationale for current larger randomized controlled trials testing the safety/efficacy profile of colchicine in COVID-19 patients.

## Introduction

Beginning in December 2019, a novel coronavirus, designated severe acute respiratory syndrome coronavirus 2 (SARS-CoV-2), has caused an international outbreak of respiratory illness termed COVID-19 [1]. The full spectrum of COVID-19 ranges from mild, self-limiting respiratory tract illness to severe progressive pneumonia, multi-organ failure, and death. Cytokines and chemokines are thought to play an important role in the severity of complications during virus infections [2]. Patients with severe COVID-19 have higher serum levels of pro-inflammatory cytokines (TNF-α, IL-1, and IL-6) and chemokines (IL-8) compared to individuals with mild disease or healthy controls, and similar levels compared to patients with Severe Acute Respiratory Syndrome (SARS) or Middle East Respiratory Syndrome (MERS) [2]. The independent association between inflammatory markers and disease severity supports the concept that abnormal inflammatory response, rather than direct viral cytopathic effects, is the main cause of the life-threatening pulmonary complications in COVID-19 patients [3]. Various mechanisms have been postulated to explain the dysregulated immune response during SARS-CoV-2 infection. In particular, the viroporin envelope (E) protein, a minor virion structural component of SARS-CoV-2, has been shown to activate the NLR family pyrin domain containing 3 (NLRP3) inflammasome, eventually causing the release of cytokines and chemokines [4, 5].

Colchicine, an old drug that has been widely used in auto-immune and inflammatory disorders [6, 7], counteracts the assembly of the NLRP3 inflammasome [8], thereby reducing the release of IL-1b and an array of other interleukins, including IL-6, that are formed in response to danger signals [7–9]. Recently, colchicine has been successfully used in two cases of life-threatening post-transplant capillary leak syndrome [10]. These patients had required mechanically ventilation and hemodialysis for weeks before receiving colchicine, which quickly restored normal respiratory function and diuresis over 48 hrs [10].

Based on this background, we started prescribing colchicine as an off-label drug in health care outpatients, and shortly after in inpatients with COVID-19 and pneumonia on lung CT scan. Herein, we report the results of an observational retrospective study in which we used inverse probability of treatment weighting based on propensity score to undergo colchicine treatment, in order to assess the hypothesis that colchicine reduces mortality and time to clinical improvement in patients with COVID-19 pneumonia.

## Patients and methods

### Patients

This is an observational, retrospective study on COVID-19 patients followed from February 25th to April 8th, 2020 at the Parma University Hospital, a tertiary health-care Centre in Parma, Italy, which was designated as a COVID-19 hub by Italian health authorities. This retrospective study included COVID-19 patients (hospitalized with pneumonia on CT scan or outpatients). We included a series of consecutive patients who received colchicine for the treatment of COVID-19 from March 1sh to April 10th, 2020. The comparison group consisted of patients that were selected by random sampling amongst those admitted at the same hospital with a diagnosis of COVID-19 and pneumonia earlier in the pandemic (from March 1st to March 18th, 2020) and who could be matched 1:1 by age (± 10 years) and sex. Because a suitable age and sex match could only be found in 59 of the 71 patients, a 1:1 match of the same sex with the closest age was obtained in 22 cases. To reduce the risk of immortal time bias (i.e. patients on colchicine cannot die before taking colchicine) patients requiring intubation in the first 24 hours after admission were excluded. Data could not be eventually extracted in one patient in the colchicine group, leaving 70 patients in the colchicine group, and 71 in the control group. The study protocol was approved by the AVEN Ethics Committee on March 31st, 2020 (prot. n. 13306).

### Criteria for COVID-19 diagnosis

We included all adult inpatients (aged ≥18 years) with a diagnosis of COVID-19 pneumonia based on: 1) CT typical findings (i.e. ground-glass opacities and/or patchy consolidation, and/or interstitial changes with a peripheral distribution), 2) positive nasopharyngeal swab test, and/or 3) serologic anti-SARS-CoV-2 antibody test. We also included health care personnel who received a diagnosis of COVID-19 based on typical symptoms and a positive nasopharyngeal swab test that were treated as outpatients and did not undergo a CT scan. As per institution protocol, all inpatients cases with clinical suspicion of COVID-19 underwent a lung CT-scan and a nasopharyngeal swab test at the time of admission. During the peak of the outbreak (i.e., at the time of patient enrollment), those that had typical clinical history and symptoms suggestive of COVID-19 along with CT findings indicative of viral pneumonia were diagnosed as COVID-19 regardless of the results of the swab test on admission. However, in those with a negative nasopharyngeal swab test at admission, diagnosis of infection was subsequently confirmed by a second nasopharyngeal swab test and/or positive test for antibodies against SARS-CoV-2.

Disease severity was quantified using a seven-point ordinal scale recommended by WHO and used by previous trials [11]. Clinical improvement was defined as a 2-point improvement on a 7-category ordinary scale which we measured every day until day 10, then at day 14, and 21) [11].

### Study drug

Colchicine was administered orally 1 mg/day from day 1 up until clinical improvement or up to a maximum of 21 days, according to physicians' preferences. As per the drug information sheet and internal guidelines, the dose was adjusted for kidney function and drug to drug interaction. Colchicine was administered orally 1 mg/day from day 1 up until clinical improvement or up to a maximum of 21 days, according to physicians' preferences. As per the internal decision, the dose was adjusted for kidney function and drug to drug interaction. The dose had to be reduced to 0.5 mg/day if the patient developed severe diarrhea. In the case of acute or chronic kidney disease with eGFR < 30mL/min not requiring dialysis, the dose was reduced to 0.5 mg/day; in

patients requiring dialysis, the dose was reduced to 0.5 mg every other day and was given after the end of each dialysis session. Patients with advanced liver impairment (up to Child-Pugh score B) could receive 0.5 mg every other day. Patients on antiretroviral drugs (including ritonavir or cobicistat) could receive one single dose of 1mg or 0.5mg according to the severity of the disease until one day from completion of antiviral treatment. Hydroxychloroquine and azithromycin did not require any dose adjustment. The decision to use anti-retroviral drugs, hydroxychloroquine, and azithromycin was up to the physician in charge of patient care.

Exclusion criteria were limited to advanced liver failure (Child-Pugh score C) and pregnancy. In particular, acute and chronic kidney failure of any degree did not represent a contraindication to the use of the drug.

### Statistical analyses

Stata release 16 (2019, StataCorp, College Station, Tx, US) was used for all the analyses. A two-tailed P-value of less than 0.05 was regarded as statistically significant. In the main analysis, we used an inverse probability of treatment weighting that was based on propensity score to construct a weighted cohort of patients who differed with respect to the use of colchicine but were similar concerning other measured characteristics. Compared to the methods of matching, inverse probability of treatment weighting allows using information from every patient, without having to exclude patients that cannot find a suitable match, which is a desirable property in the presence of sparse data. To calculate the inverse probability of treatment weights, we estimated each patient's propensity to undergo colchicine treatment, using a logistic regression model that included predictor variables that had been selected based on their a priori possibility of confounding the relationship between colchicine use and mortality (age, sex, categorical variate indicating the severity of conditions at onset namely, non-hospitalized, hospitalized without oxygen, hospitalized and requiring supplemental oxygen, hospitalized requiring non-invasive ventilation, shortness of breath, cough, history of diabetes, history of hypertension, history of cancer, use of antibiotics, use of anti-retroviral drugs, use of hydroxychloroquine, use of i.v. steroids, use of tocilizumab). We assigned patients who received colchicine a weight of 1÷(propensity score) and those who did not receive colchicine a weight of 1÷(1−propensity score). To reduce the variability in the inverse probability of treatment–weighted models, we used stabilized weights [12]. Hazard-ratios (under the assumption of the absence of unmeasured confounding) that are estimated by propensity-score methods are more like the effects estimated in a randomized, controlled trial than those estimated by multivariable Cox regression [12]. We compared the distributions of categorical variables using the chi-square test in the unweighted cohort and the weighted logistic-regression models in the weighted cohort; of continuous variable by Mann-Whitney test. In the propensity-score–weighted cohort, we compared cumulative mortality between the colchicine group and the control group by plotting weighted survival functions and by estimating the hazard ratio for death associated with the use of colchicine with weighted Cox proportional-hazards models [12]. Change in lab parameters was estimated by fitting weighted random coefficients models via maximum likelihood which allowed for lack of consistency among subjects in the timing of the lab measurements. For the purpose of the analysis, C-reactive protein was log base 2 transformed (on unit decrease represent halving of C-reactive protein values).

## Results

### Patients

We included 141 consecutive patients with COVID 19. All patients were followed-up until death or 21 days after admission. In the 70 patients taking colchicine, the median time from

hospital admission to first colchicine administration was 4 days (interquartile range: 2 to 9), the median number of days on colchicine treatment was 6 (interquartile range: 2 to 13), with ten patients taking a single dose because of concomitant anti-protease treatment. Fiftynine patients (84%) started with 1 mg/day and eleven patients started with 0.5mg daily because of drug-to-drug interaction (eleven patients) and/or chronic renal failure (nine patients). **Table 1** and **S1 Table** summarize selected demographic and clinical characteristics of the study population before and after propensity-score weighting, respectively. Most baseline demographics and clinical characteristics were similar between the two groups. However, in the unweighted cohort, patients who received colchicine had, at admission, more often dyspnea, received more often supplemental oxygen and non- invasive mechanical ventilation (**Table 1**); a higher percentage of them were on antibiotics or antivirals at enrollment (**Table 1**); because of the higher disease severity in the colchicine group, more patients in the colchicine group received tocilizumab [10] (14.3%) in the colchicine vs. 4 (5.6%) in the control group; P = 0.086]. (**Table 1**). Those characteristics were balanced in the propensity-weighted cohort (**S1 Table**). In the propensity-weighted cohort, none of the characteristics statistically differed between the groups (**S1 Table**), and all the standardized differences were less than 20% (median +1.3%; range: -17.9 to +19.1%; interquartile range: -12.5 to +6.2%).

## Outcomes

In the propensity-weighted cohort, the 21-day crude cumulative incidence of death was 7.5% In the colchicine group and 28.5% in the control group (P = 0.006; adjusted hazard ratio of death: 0.24 [95%CI: 0.09 to 0.67]) (**Fig 1**); 21-day clinical improvement occurred in 40.0% of the patients on colchicine and in 26.6% of control patients (P = 0.048); adjusted relative improvement rate: 1.80 (95%CI: 1.00 to 3.22). In exploratory subgroup analyses, the relative hazard reduction associated with colchicine was evident across different patient categories (**Fig 2**). However, in some categories, the hazard ratio could not be reliably estimated because of sparse data within some strata (i.e. patients with cancer, CKD, and receiving tocilizumab), and of no deaths (outpatients).

The strong association between the use of colchicine and reduced mortality was further supported in hospitalized patients by the diverging linear trends of log2 C-reactive protein levels and lymphocyte count since admission. In fact, log2 C-reactive protein had a sharper decrease in the colchicine group compared to the control group (P = 0.009, **Fig 3A**), a log reduction of 1 is equivalent to a halving of the concentration of C-reactive protein, in mg/L) while lymphocyte count showed a sharper increase in lymphocyte count in the colchicine group compared to the control group (P = 0.018) (**Fig 3B**). The improvement in lymphocyte count in the colchicine group compared to the control group was mirrored by a sharper improvement in the neutrophil-to-lymphocyte ratio (P = 0.003). Unlike patients in the control group, those who received colchicine had IL-6 serially measured because of the physicians' expectation that the drug would decrease IL-6 levels. Over the first week after admission, Log2 IL-6 levels decreased by almost two log units (P<0.001 for linear trend), implying a decrease to almost 25% of baseline levels (**Fig 4**). Because of the retrospective nature of the study, we did not have enough IL-6 measurements in the control group to make a reliable comparison.

## Safety

Colchicine was well tolerated. Only four (7%) of the patients had to withdraw the drug because of side effects, two because of diarrhea, and two for skin rash.

**Table 1. Demographic and clinical characteristics of the study population before propensity-score weighting.**

| | Control | Colchicine | P value* |
|---|---|---|---|
| | (N = 71) | (N = 70) | |
| **Age—yrs** | 62.5 (14.5) | 60.5 (13.4) | 0.39 |
| **Male gender** | 49 (69.0) | 51 (72.9) | 0.62 |
| *Comorbidities* | | | |
| **Diabetes** | 13 (18.3) | 11 (15.7) | 0.68 |
| **Cancer** | 6 (8.4) | 4 (5.7) | 0.53 |
| **Hypertension** | 43 (60.6) | 39 (55.7) | 0.56 |
| **CKD** | 13 (18.3) | 15 (21.4) | 0.64 |
| **BMI**§ | 25.0 (5.5) | 25.9 (5.4) | 0.54 |
| *Disease severity at diagnosis* | | | |
| **Non-hospitalized** | 13 (18.3) | 10 (14.3) | 0.004 |
| **Hosp. w/o O₂** | 24 (33.8) | 8 (11.4) | |
| **Hosp. with O₂** | 24 (33.8) | 31 (40.3) | |
| **Hosp. with NIV** | 10 (14.1) | 21 (30.0) | |
| *Clinical characteristics at diagnosis* | | | |
| **Fever** | 70 (98.6) | 68 (97.1) | 0.55 |
| **Dyspnea** | 26 (36.6) | 50 (71.4) | <0.001 |
| **Cough** | 50 (70.4) | 43 (61.4) | 0.26 |
| **Arthro-myalgias** | 6 (8.4) | 12 (17.1) | 0.12 |
| **Diarrhea** | 5 (7.0) | 9 (12.9) | 0.25 |
| *Lab values at hospital admission (Inpatients)* | | | |
| **CRP- mg/dL** | 115.2 (4,>250) | 116.6 (13,>250) | 0.59 |
| | n = 41 | n = 54 | |
| **IL-6 –pg/mL** | NA | 127.6 (0.1, 860) | NA |
| | n = 0 | n = 33 | |
| **PCT–ng/mL** | 0.17 (0,1.9) | 0.17 (0,3) | 0.99 |
| | n = 26 | n = 39 | |
| **Lymph.–N/mm³** | 966 (532) | 1072 (539) | 0.33 |
| | n = 45 | n = 57 | |
| **Neut/Lymph ratio** | 5.4 (1.3,37) | 5.2 (1.3,34) | 0.62 |
| | n = 22 | n = 36 | |
| **D-dimer–ng/mL** | 869 (164,>9000) | 1103(238,>9000) | 0.79 |
| | n = 25 | n = 44 | |
| **Ferritin - µg/L** | 824(90,2594) | NA | NA |
| | n = 14 | | |
| **sCreatinine–mg/dL** | 0.8 (0.5,7.2) | 0.8 (0.4,21.0) | 0.93 |
| | n = 44 | n = 54 | |
| *COVID-19 Treatment* | | | |
| **Use of antibiotics** | 39 (54.9) | 60 (85.7) | <0.001 |
| **Antiviral treatment** | 45 (63.4) | 40 (57.1) | 0.45 |
| **Hydroxychloroquine** | 46 (64.8) | 53 (75.7) | 0.16 |
| **i.v. steroids** | 9 (12.7) | 17 (24.3) | 0.076 |
| **Tocilizumab** | 4 (5.6) | 10 (14.3) | 0.086 |

Categorical data are expressed as number (%), continuous data as average (SD), or median (range).

Lab values had missing data, therefore the number of available data is reported for each group.

*The P value for categorical variables was calculated by the chi-square test, for continuous variables by Mann-Whitney test.

§BMI was available in 70 of the 141 patients.

CKD, chronic kidney disease; NIV, non-invasive mechanical ventilation; Hosp., hospitalized; O₂, supplemental oxygen; CRP, C-reactive protein; PCT, procalcitonin; Lymph., Lymphocyte count; sCreatinine, serum creatinine, PTL, platelet. Neut/Lymph ratio, neutrophil to lymphocyte ratio, antiviral treatment, anti-retroviral drugs (lopinavir or lopinavir + ritonavir or cobicistat).

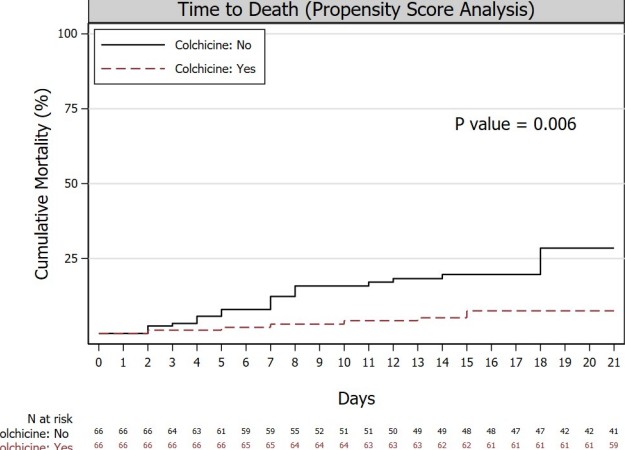

**Fig 1. Time to death.** Cumulative incidence of death since hospital admission (inpatients) or diagnosis (outpatients) in the two treatment groups. The 21-day crude cumulative incidence of death was 7.5% In the colchicine group and 28.5% in the control group (P = 0.006; adjusted hazard ratio of death: 0.24 [95%CI: 0.09 to 0.67]). The cumulative incidence curves and number at risk at the bottom of the figure refer to the cohort after inverse probability of treatment weighting (numbers at risk are rounded to the nearest integer).

## Discussion

In this observational retrospective study, using propensity score analysis, we found that colchicine administration was associated with a significant reduction in mortality and accelerated clinical improvement in patients with COVID-19.

The use of colchicine in COVID-19 has a sound biological rationale. The drug, which is well-known for its immunomodulatory properties in severe auto-inflammatory diseases [6], has been

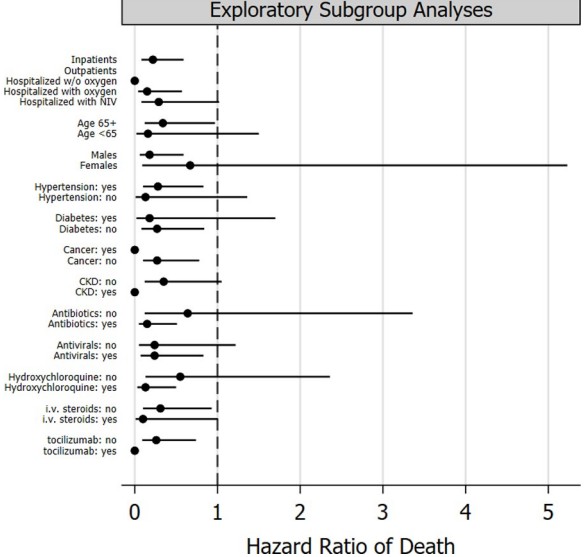

**Fig 2. Exploratory subgroup analyses.** Exploratory subgroup performed by Cox regression analyses adjusted by inverse probability of treatment weighting. The relative hazard reduction associated with colchicine was evident across different patient categories. However, in some categories the hazard ratio could not be reliably estimated because of sparse data (i.e. patients with cancer, with CKD, and receiving tocilizumab), and of no deaths in the stratum (outpatients). Solid circles represent the hazard ratio of death. Vertical bars represent 95 percent confidence intervals.

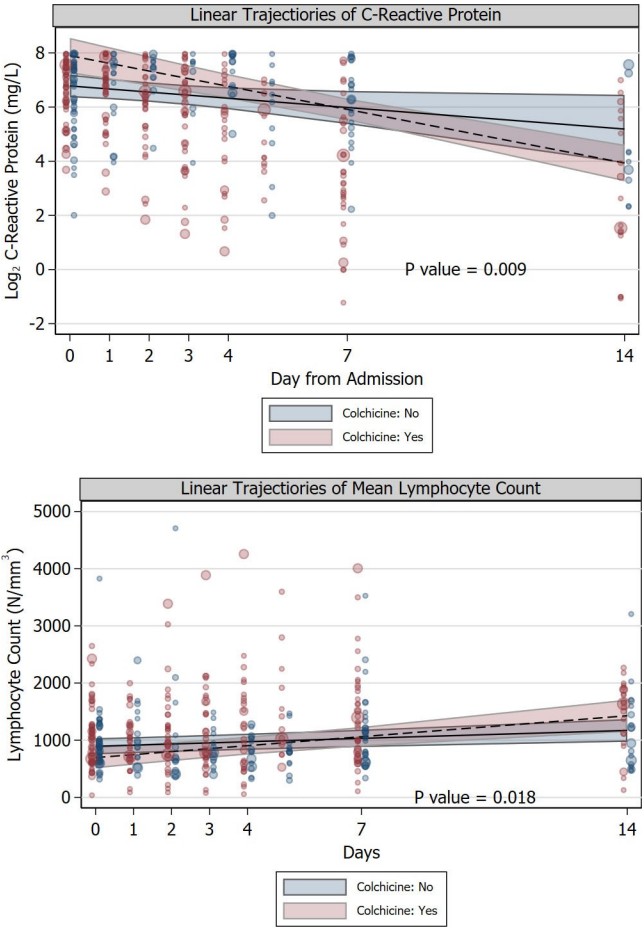

**Fig 3.** A-B. Linear trajectories since hospital admission of the mean of C-reactive protein and of lymphocyte count. Linear trajectories since hospital admission of the mean of C-reactive protein (Panel A), and of lymphocyte count (Panel B), in patients who received colchicine (red) or did not receive colchicine (blue). Trajectories express the average linear trends predicted from the propensity score-adjusted random coefficients model (see text). C-reactive protein is expressed as logarithm base 2, therefore1-unitdecrease implies halving of C-reactive protein levels. Log2 C-reactive protein had a sharper decrease in the colchicine group compared to the control group (P = 0.009); lymphocyte count showed a sharper increase in lymphocyte count in the colchicine group compared to the control group (P = 0.018). Bands represent 95% confidence intervals. Data values are indicated by circles the diameter of which is proportional to the inverse probability of treatment weight based on the propensity score that was used in the all the analyses.

recently shown to modulate the release of circulating cytokines such as IL-6 [8, 9]. In addition, colchicine reduces lung injury in experimental acute respiratory distress syndrome (ARDS) [13]. Finally, colchicine may be an effective treatment for inflammation-induced thrombosis [14]. At variance with anti-IL6/IL-6 receptor antagonists, colchicine is widely available on the market, and inexpensive [6], and it is, therefore, suitable for massive use in countries where biologics are not easily available. We contend that colchicine use may be particularly attractive as a treatment to prevent the rapid progression of mild and moderate forms of COVID-19 to severe forms of respiratory failure and likewise it would be a useful drug also to treat a possible resurgence of SARS and MERS. Our study confirms the well-known safe toxicity profile of colchicine since the drug was withdrawn because of side effects (all of minor severity) in only 7% of treated patients.

Due to its retrospective nature, several baseline laboratory values and respiratory parameters were missing, therefore we could not fully adjust for all baseline differences between the

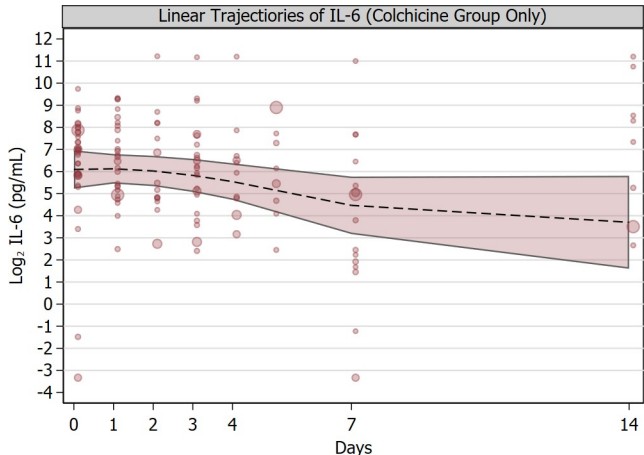

**Fig 4. Linear trajectories of IL-6.** Mean trajectory since hospital admission of log IL-6 serum levels (Panel A) in hospitalized patients who received colchicine. The dotted line is estimated based on random-coefficients mixed model with time included as polynomial variable.IL-6 serum level is expressed as logarithm base 2, therefore 1-unit decrease implie shalving of IL-6 serum levels. Bands represent 95% confidence intervals. Data values are indicated by circles the diameter of which is proportional to the inverse probability of treatment weight based on propensity score which was used in the analysis for comparison between treatment groups. The linear decreasing trend was statistically significant (P<0.001).

two groups in the propensity score analyses. Nonetheless, patients receiving colchicine had, on average, more severe symptoms such as dyspnea compared to patients in the control group. In fact, because of the worse baseline conditions, patients on colchicine tended to receive more frequently tocilizumab and i.v. steroids. However, in our subgroup analyses (**Fig 2**) the effect of colchicine was also evident after excluding patients who received tocilizumab or i.v. steroids. During the study period, no other anti-inflammatory agent besides steroids, was used as for the standard of care at that time.

We cannot, however, exclude that differences in mortality between groups may be related to an underlying decreasing trend in mortality rates that might have occurred during the study period because of improved patient management. However, it is unlikely that an underlying trend accounts for all estimated effect of colchicine since the study period of the control and treatment group were very close to each other. Moreover, the timing of colchicine start and treatment duration was heterogeneous between patients in the colchicine group, the duration of treatment was often short because of the frequent use of protease-inhibitors-boosted anti-retroviral drugs, and colchicine was early withdrawn for an adverse event in 7% of patients. However, this study drawback should have biased our findings toward the null. Finally, although the selection of patients who could survive enough to receive colchicine may have caused an overestimation of the colchicine effect because of immortal time bias, the cumulative mortality curves diverged throughout the follow-up (**Fig 1**).

To the best of our knowledge, there is only one small randomized study [12], one observational study, which showed remarkably similar findings to our study [15], and two small case series of outpatients on colchicine in COVID-19 patients [16, 17], all showing clinical benefit. In the randomized study [12], which was performed in Greece, patient recruitment was terminated because of slow enrollment as a result of the rapid flattening of the curve of COVID-19. In that study, the clinical endpoint was defined as a 2-grade increase on an ordinal clinical scale that we used in our study, within a time frame of 21 days. The maintenance dosage of colchicine was the same as in our study. Of the 180 originally planned based on the clinical endpoint, only 105 patients were enrolled. Nonetheless, the clinical primary endpoint rate was 14.0% in the

control group (7 of 50 patients) and 1.8% in the colchicine group (1 of 55 patients) (odds ratio, 0.11; 95% CI, 0.01–0.96; P = 0.02), a finding which is fully consistent with the results of our retrospective observational study. The only large randomized study that so far showed a clear reduction in mortality among COVID-19 patients used dexamethasone [18]. Similar to colchicine, dexamethasone is another anti-inflammatory drug, which further supports the biological plausibility of our study finding [18]. In fact, early use of anti-inflammatory drugs may help to prevent complications such as pulmonary fibrosis, and thromboembolic disease.

In our study, the subgroup of health care personnel who did not require supplemental oxygen and were treated as outpatients had apparently the greatest benefit (**Fig 2**). These findings might imply that colchicine is most effective in the early stages of innate immune response. However, due to the small sample size, we cannot draw definite conclusions.

Our findings need to be confirmed by properly designed prospective trials. Indeed, we and others have started enrolling patients in multicenter randomized clinical trials on colchicine treatment for COVID-19. At the time of writing, twelve randomized studies on colchicine are registered on ClinicalTrial.gov. Some of these trials may eventually fail to provide results because, by the time the trials have started enrolling patients in a given country, the outbreak is declining and enrollment of patients may become difficult. Therefore, evidence coming from non-randomized studies may be fundamental to guide COVID-19 patient treatment.

In conclusion, pending the results from randomized control trials, our retrospective study provides evidence that colchicine may be a safe and effective drug for the treatment of COVID-19.

## Supporting information

**S1 Table. Demographic and clinical characteristics of the study population after propensity-score weighting.**
(DOCX)

**S1 Dataset.**
(XLS)

**S1 File. Stata code main analysis.**
(TXT)

## Author Contributions

**Conceptualization:** Lucio Manenti, Umberto Maggiore, Paolo Cravedi, Licia Peruzzi.

**Data curation:** Lucio Manenti.

**Formal analysis:** Lucio Manenti.

**Writing – original draft:** Lucio Manenti, Umberto Maggiore, Paolo Cravedi.

**Writing – review & editing:** Umberto Maggiore, Enrico Fiaccadori, Tiziana Meschi, Anna Degli Antoni, Antonio Nouvenne, Andrea Ticinesi, Nicoletta Cerundolo, Beatrice Prati, Marco Delsante, Ilaria Gandoflini, Lorenzo Donghi, Micaela Gentile, Maria Teresa Farina, Vincenzo Oliva, Cristina Zambrano, Giuseppe Regolisti, Alessandra Palmisano, Caterina Caminiti, Enrico Cocchi, Carlo Ferrari, Leonardo V. Riella, Paolo Cravedi, Licia Peruzzi.

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
