## [Decision Letter · Decision Letter 0]

14 Jan 2021

PONE-D-20-29570

Reduced mortality in COVID-19 patients treated with colchicine: results from a retrospective, observational study

PLOS ONE

Dear Dr. Cravedi,

Thank you for submitting your manuscript to PLOS ONE. After careful consideration, we feel that it has merit but does not fully meet PLOS ONE’s publication criteria as it currently stands. Therefore, we invite you to submit a revised version of the manuscript that addresses the points raised during the review process.

We look forward to receiving your revised manuscript.

Kind regards,

Antonio Cannatà

Academic Editor

PLOS ONE

2. Please note that all PLOS journals ask authors to adhere to our policies for sharing of data and materials: https://journals.plos.org/plosone/s/data-availability. According to PLOS ONE’s Data Availability policy, we require that the minimal dataset underlying results reported in the submission must be made immediately and freely available at the time of publication. As such, please remove any instances of 'unpublished data' or 'data not shown' in your manuscript and replace these with either the relevant data (in the form of additional figures, tables or descriptive text, as appropriate), a citation to where the data can be found, or remove altogether any statements supported by data not presented in the manuscript.

Furthermore, this is a retrospective study. As such, we do not feel that any conclusions on the intervention effects can be supported; as such, we ask that you revise the text to avoid unsupported statements.

4. Please include your tables as part of your main manuscript and remove the individual files. Please note that supplementary tables (should remain/ be uploaded) as separate "supporting information" files

Additional Editor Comments:

Editor:

Please, in addition to reviewers' comments, please reply to the following issues:

- A sensitivity analysis excluding patients using other ant-inflammatory drugs should be performed.

- Please discuss more in details the findings regarding the effect of colchicine in patients not requiring O2 therapy.

- the introduction is too wordy. Please consider shortening it.

- please provide the mean dose of colchicine and describe how many patients received 0.5 mg vs 1 mg. There was any difference in outcomes?

Reviewers' comments:

Reviewer's Responses to Questions

**Comments to the Author**

1. Is the manuscript technically sound, and do the data support the conclusions?

Reviewer #1: Yes

2. Has the statistical analysis been performed appropriately and rigorously? 

Reviewer #1: Yes

3. Have the authors made all data underlying the findings in their manuscript fully available?

Reviewer #1: No

4. Is the manuscript presented in an intelligible fashion and written in standard English?

Reviewer #1: Yes

5. Review Comments to the Author

Reviewer #1: This article can provide more information about the effect of colchicine on the treatments of COVID-19 patients, however we want to make it better by some recommendations.

Abstract

Please add the number of patients(both groups) in method.

method

please explain more about the severity of diseases in both groups? whether the severity of the patients on colchicine were higher than control group?

Please explain more about the history of other drugs or supplements consumption in both groups.

Did patients take any drug with colchicine Simultaneously which may effect the result?

There is not any information about the weight and Body Mass Index of patients which can effect the results( since obesity is a risk factor for covid-19 patients). It will be better to add the Antropometric measurements if they are available?

Discussion

Please explain more that how the colchicine may reduce the death in COVID-19 patients according to documents?

6. PLOS authors have the option to publish the peer review history of their article (what does this mean?). If published, this will include your full peer review and any attached files.

Reviewer #1: No

---

## [Author Response · Author response to Decision Letter 0]

15 Jan 2021

ID: PONE-D-20-29570

Title: Reduced mortality in COVID-19 patients treated with colchicine: results from a retrospective, observational study

Rebuttal letter 

RE: We modified the manuscript to meet PLOS ONE’s style requirements.

2. Please note that all PLOS journals ask authors to adhere to our policies for sharing of data and materials: https://journals.plos.org/plosone/s/data-availability. According to PLOS ONE’s Data Availability policy, we require that the minimal dataset underlying results reported in the submission must be made immediately and freely available at the time of publication. As such, please remove any instances of 'unpublished data' or 'data not shown' in your manuscript and replace these with either the relevant data (in the form of additional figures, tables or descriptive text, as appropriate), a citation to where the data can be found, or remove altogether any statements supported by data not presented in the manuscript.

RE: We have reported the data in the amended version of the results.

Furthermore, this is a retrospective study. As such, we do not feel that any conclusions on the intervention effects can be supported; as such, we ask that you revise the text to avoid unsupported statements.

RE: We modified the abstract conclusions accordingly. We verified that the rest of the manuscript does not include any statement implying causation.

RE: We have reported the relevant data in the amended results section of the manuscript.

4. Please include your tables as part of your main manuscript and remove the individual files. Please note that supplementary tables (should remain/ be uploaded) as separate "supporting information" files

RE: Tables are now embedded in the manuscript. 

RE: We listed the supporting Information at the end of the manuscript.

Editor Comments

1. A sensitivity analysis excluding patients using other ant-inflammatory drugs should be performed.

RE: During the study period no anti-inflammatory agent besides steroids, colchicine and tocilizumab, was used as for the standard of care at that time. We specified this issue in the amended version. On the other hand, sensitivity analysis for steroids and tocilizumab is reported in Fig 2.

2. Please discuss more in details the findings regarding the effect of colchicine in patients not requiring O2 therapy.

RE: We added a comment on this in the amended version of the discussion section.

3. The introduction is too wordy. Please consider shortening it.

RE: We shortened the introduction according to the Editor’s advice.

4. please provide the mean dose of colchicine and describe how many patients received 0.5 mg vs 1 mg. There was any difference in outcomes?

RE: In the amended version of the results section we specified that the colchicine starting dose of 0.5mg was reserved to patients with drug-to-drug-interactions and/or chronic kidney failure in whom 0.5mg daily correspond to 1mg daily or more. With the limitations of different baseline comorbidities, we did not detect any sign that 0.5mg was associated with increased mortality (1 death out of the eleven patients, i.e. 9%) 

We thank the Editor for the helpful comments that helped us improving the manuscript.

Reviewers' comments:

1. Abstract: Please add the number of patients (both groups) in method.

RE: We included the number of patients of the unweighted cohort in the revised abstract.

2. Please explain more about the severity of diseases in both groups? whether the severity of the patients on colchicine were higher than control group?

RE: We added a comment on the difference in the unweighted cohort in the amended version of the results.

3. Please explain more about the history of other drugs or supplements consumption in both groups. Did patients take any drug with colchicine Simultaneously which may affect the result?

RE: During the study period, no anti-inflammatory agent besides steroids, colchicine, and tocilizumab, were used as for the standard-of-care at that time. We specified this issue in the amended version. Sensitivity analysis for steroids and tocilizumab is reported in Fig 2.

4. There is not any information about the weight and Body Mass Index of patients which can affect the results (since obesity is a risk factor for covid-19 patients). It will be better to add the Antropometric measurements if they are available?

RE: Unfortunately, height was not available in a substantial proportion of patients. Therefore, in these patients, BMI was not available. However, we added available BMI data in the amended version of Table 1. There was no significant difference between groups.

5. Discussion: please explain more that how the colchicine may reduce the death in COVID-19 patients according to documents?

RE: Revised version of the introduction and discussion include statements on the potential mechanisms responsible for the beneficial effects of colchicine on COVID-19 associated mortality.

We thank the Reviewer for the constructive feedback that helped improving the quality of the manuscript.

---

## [Decision Letter · Decision Letter 1]

5 Feb 2021

PONE-D-20-29570R1

Reduced mortality in COVID-19 patients treated with colchicine: results from a retrospective, observational study

PLOS ONE

Dear Dr. Cravedi,

Thank you for submitting your manuscript to PLOS ONE. After careful consideration, we feel that it has merit but does not fully meet PLOS ONE’s publication criteria as it currently stands. Therefore, we invite you to submit a revised version of the manuscript that addresses the points raised during the review process.

Please, make sure the limitation section properly describes the potential bias of this study.

We look forward to receiving your revised manuscript.

Kind regards,

Antonio Cannatà

Academic Editor

PLOS ONE

Reviewers' comments:

Reviewer's Responses to Questions

**Comments to the Author**

1. If the authors have adequately addressed your comments raised in a previous round of review and you feel that this manuscript is now acceptable for publication, you may indicate that here to bypass the “Comments to the Author” section, enter your conflict of interest statement in the “Confidential to Editor” section, and submit your "Accept" recommendation.

Reviewer #1: All comments have been addressed

Reviewer #2: (No Response)

2. Is the manuscript technically sound, and do the data support the conclusions?

Reviewer #1: Yes

Reviewer #2: Yes

3. Has the statistical analysis been performed appropriately and rigorously? 

Reviewer #1: Yes

Reviewer #2: Yes

4. Have the authors made all data underlying the findings in their manuscript fully available?

Reviewer #1: Yes

Reviewer #2: Yes

5. Is the manuscript presented in an intelligible fashion and written in standard English?

Reviewer #1: Yes

Reviewer #2: Yes

6. Review Comments to the Author

Reviewer #1: Thank you so much for your revision. Now, the results of this paper can be more useful in future trial study.

Reviewer #2: Manenti et al. in their paper report an interesting retrospective analysis exploring the prognostic effect of an antinflammatory drug widely used in other scenarios: colchicine. The authors found that after propensity score matching the treatment with colchicine was associated with a reduced incidence of death and a significant improvement. The paper is of interest, considering also the recent relevance gathered by colchicine in COVID-19. Moreover, the paper is methodically well conducted and well written. However, I have some concern:

Major revisions required

The needing to discontinue colchicne in 7% of patients introduces a bias, due to a shorter duration of the treatment. Therefore, if it is not possible to replace these patients with other similarly matched patients, this limitation should be acknowledged.

A sensitivity analysis to make sure that a selection bias was avoided would increase the quality of the paper.

The match is based only on gender and age. However, in COVID infection many other factors increase the risk of adverse outcome (i.e. comorbidities). The population should be matched also for comorbidities (at least the number of comorbidities) and the COVID treatment. To overcome this limitation, I would suggest to adjust the HR of the colchicine treatment for the variables with a statistically significant difference in the two groups (i.e. disease severity, COVID treatment and dyspnoea. Otherwise this limitation should be stated.

The authors state that patients with a clinical scenario of COVID infection were included in the analysis even if the first swab was negative, if then the second swab was positive. In which ward were these patients admitted? If they were admitted to a COVID unit it is likely that the result of the second swab was affected by the presence of the virus in the ward.

Minor revision required

Please codify all the abbreviation (i.e. SARS-CoV2, SARS and MERS are never stated in full).

The authors state that increased serum concentration of cytochines are found in SARS and MERS patients as well. I would suggest to discuss the evidences supporting the use of colchicine in SARS and MERS.

In the methods section authors state that patients included in the current analysis were admitted at Parma University Hospital, but a share of patients were outpatients. Please clarify.

The treatment of acute pericarditis requires a colchicine dose reduction to 0.5 mg/day in case of body weight < 70 Kg. Was this dose reduction considered?

According to table 1, 33 patients had IL-6 dosed at admission but it is not reported the average value.

Colchicine may increase the risk of bacterial pneumonia. In this view, even though I admit that the diagnosis is very challenging in this scenario, authors should state if any patient had a suspicion of bacterial pneumonia at baseline during the study follow-up.

7. PLOS authors have the option to publish the peer review history of their article (what does this mean?). If published, this will include your full peer review and any attached files.

Reviewer #1: **Yes: **Sara Asadi

Reviewer #2: No

---

## [Author Response · Author response to Decision Letter 1]

10 Feb 2021

ID: PONE-D-20-29570

Title: Reduced mortality in COVID-19 patients treated with colchicine: results from a retrospective, observational study

Rebuttal letter 

Reviewer #2: 

Manenti et al. in their paper report an interesting retrospective analysis exploring the prognostic effect of an antinflammatory drug widely used in other scenarios: colchicine. The authors found that after propensity score matching the treatment with colchicine was associated with a reduced incidence of death and a significant improvement. The paper is of interest, considering also the recent relevance gathered by colchicine in COVID-19. Moreover, the paper is methodically well conducted and well written. 

Response: We thank the Reviewer for the positive comments.

The needing to discontinue colchicne in 7% of patients introduces a bias, due to a shorter duration of the treatment. Therefore, if it is not possible to replace these patients with other similarly matched patients, this limitation should be acknowledged.

Response: We acknowledge this issue in the study limitations (line 338). We also explained that “this study drawback should have biased our findings toward the null.”

A sensitivity analysis to make sure that a selection bias was avoided would increase the quality of the paper.

Response: The reviewer raised an important issues that we addressed in Figure 2 when we report the results after excluding each category based on severity of disease, comorbidity, concomitant treatment.

The match is based only on gender and age. However, in COVID infection many other factors increase the risk of adverse outcome (i.e. comorbidities). The population should be matched also for comorbidities (at least the number of comorbidities) and the COVID treatment. To overcome this limitation, I would suggest to adjust the HR of the colchicine treatment for the variables with a statistically significant difference in the two groups (i.e. disease severity, COVID treatment and dyspnoea. Otherwise this limitation should be stated.

Response: The reviewer raised a relevant issue that we faced (similarly to previous retrospective published study such as N Engl J Med 2020; 382:2411-2418) using the state-of-the-art methodology to control for confounding in treatment comparison in observational studies, namely inverse probability of treatment weighting based on propensity score to construct a weighted cohort of patients (see Statistical methods Lines165-187). The analysis adjusted for imbalances between groups in baseline severity, such as chronic kidney diabetes, diabetes, cancer, hypertension, and concomitant drugs. Matching for age and sex was exclusively used for sampling controls. The statistician provided the code for the main analyses as supplementary material (below).

*-------------------------------------------------------------------------------

*Start analyses with inverse probability of treatment propensity-score weighting 

*-------------------------------------------------------------------------------

*----------------------------------------------------+

* Start Calculating IPTW andChecking IPTW Assumptions

*----------------------------------------------------+

use COLCHICINE_DATASET, clear

#delimit ;

global ps_ivar "c.age i.SEX 

 i.grp_severity1 i.grp_severity2 i.grp_severity3 i.grp_severity4 

 i.hist_ckd i.hist_diabetes i.hist_cancer i.hist_hypertension 

 i.hist_dyspnea i.hist_cough i.hist_arhtrmyalg i.hist_diarrhea 

 i.hist_antibiotics_done i.antiviral i.hydroxyxhloroquine 

 i.steroids i.toci_yes";

#delimit cr

#delimit ;

global ps_nivar "age SEX 

 grp_severity1 grp_severity2 grp_severity3 grp_severity4 

 hist_ckd hist_diabetes hist_cancer hist_hypertension 

 hist_dyspnea hist_cough hist_arhtrmyalg hist_diarrhea 

 hist_antibiotics_done antiviral hydroxyxhloroquine 

 steroids toci_yes";

#delimit cr

logit colch_tp $ps_ivar 

* calculate stabilized inverse probability of treatment weights 

* (Austin Stat Med 2016; 35: 5642)

predict pr1, pr

gen pr0 = 1 - pr1

qui summ pr1, meanonly

local mpr1 = r(mean)

qui summ pr0, meanonly

local mpr0 = r(mean)

gen stab_ipw = .

replace stab_ipw = `mpr1' * 1/pr1 if colch_tp ==1

replace stab_ipw = `mpr0' * 1/pr0 if colch_tp ==0

drop if missing(stab_ipw)

// check common support

summ stab_ipw if colch_tp == 0

tw kdensity stab_ipw if colch_tp == 0, lcolor(black) lpattern(solid) || ///

 kdensity stab_ipw if colch_tp == 1, lcolor(maroon) lpattern(dash) || ///

 , ///

 ytitle("Density") ylabel(, angle(horizontal) format(%3.1f)) ///

 xlabel(, forma(%3.1f)) ///

 xtitle("Stabilized inverse probability of treatment weights") ///

 legend(pos(2) ring(0) rows(2) order(1 "Control" 2 "Colchicine")) ///

 scheme(s1mono)

/// Start computation of Standardized differences

teffects ipw (death) (colch_tp $ps_ivar, logit)

tebalance summarize

foreach var of varlist ///

 $ps_nivar {

 di _newline(3) _col(8) in gr "------------------ Standardized difference of var: `var'"

 qui summ `var' [aw = stab_ipw] if colch_tp == 0

 local m0 = r(mean)

 local v0 = r(Var)

 qui summ `var' [aw = stab_ipw] if colch_tp == 1

 local m1 = r(mean)

 local v1 = r(Var)

 local std = (`m0' - `m1') / sqrt((`v0' + `v1')/2)

 di _col(8) in ye %3.1f `std' * 100 " %" 

 }

/// End computation of standardized differences

*-----------------------------+

* End Check IPTW Assumptions

*-----------------------------+

*---------------------------------------------------------------+

* Start Survival Analysis (21-day mortality)

*---------------------------------------------------------------+

stset FOLLOW_DATE [pw = stab_ipw], id(id) origin(admission_date) fail(death == 1) ///

 exit(time admission_date + 21)

stcox colch_tp , cluster(id)

qui test _b[colch_tp] = 0

local pval = r(p) 

local spval = string(`pval', "%4.3f")

/// make the plot

#delimit ;

global kmstuff "risktable(, title("N at risk", size(*.7))) 

 risktable(, color("black") group(#1) size(*.7)) 

 risktable(, color("maroon") group(#2) size(*.7)) 

 risktable(, rowtitle("Colchicine: No ") group(#1) size(*.7)) 

 risktable(, rowtitle("Colchicine: Yes") group(#2) size(*.7))

 plot1opts(lcolor("black") lwidth(*1.0) lpattern(solid)) 

 plot2opts(lcolor("maroon") lwidth(*1.0) lpattern(dash)) 

 legend(cols(1) position(11) rows(3) ring(0) size(*.7)

 lstyle(none) 

 order(1 "Colchicine: No" 2 

 "Colchicine: Yes")) 

 ysc(range(.75 1)) 

 ylab(.25 "25" .5 "50" .75 "75" 1 "100", 

 angle(horizontal) labsize(*.7) grid ) 

 xsc(titlegap(5)) xlab(0(1)21, format(%3.0f) labsize(*.7))";

#delimit cr

sts graph, by(colch_tp) failure ///

 $kmstuff risktable(, format(%3.0f)) ///

 xtitle("Days") ///

 ytitle("Cumulative Mortality (%)") scheme(s1mono) ///

 title("Time to Death (Propensity Score Analysis)", size(*0.8) ///

 box bexpand bmargin(0 0 0 0)) ///

 text(.7 17 "P value = `spval'")

graph export colch_propensity_score_iptw_death_only.png, replace

*-------------------------------------------------------------------+

* End Survival Analysis (21-day mortality)

*-------------------------------------------------------------------+

The authors state that patients with a clinical scenario of COVID infection were included in the analysis even if the first swab was negative, if then the second swab was positive. In which ward were these patients admitted? If they were admitted to a COVID unit it is likely that the result of the second swab was affected by the presence of the virus in the ward.

Response: We understand the Reviewer’s concern. However, false-negative RT-PCR test from swab tests have been well documented. If initial testing was negative, COVID-19 diagnosis was made based on the presence interstitial pneumonia and then confirmed in the next days with a repeated swab test. Therefore, the possibility that the pneumonia was due to a subsequent SARS-CoV2 infection is very unlikely.

Please codify all the abbreviation (i.e. SARS-CoV2, SARS and MERS are never stated in full).

Response: According to the Reviewer’s suggestion, we codified all the abbreviations.

The authors state that increased serum concentration of cytochines are found in SARS and MERS patients as well. I would suggest to discuss the evidences supporting the use of colchicine in SARS and MERS.

Response: In the revised discussion, we included a sentence specifying that “likewise colchicine would be a useful drug also to treat a possible resurgence of SARS and MERS.”

In the methods section authors state that patients included in the current analysis were admitted at Parma University Hospital, but a share of patients were outpatients. Please clarify.

Response: We thank the Reviewer for pointing this out. In the revised manuscript, we clarified that patients where were followed-up (Line 110). In Line 113, we described the study cohort more in detail.

The treatment of acute pericarditis requires a colchicine dose reduction to 0.5 mg/day in case of body weight < 70 Kg. Was this dose reduction considered?

Response: We considered a dose reduction for liver or chronic kidney disease. We adopted as reference dose the one used in autoinflammatory disease, like Mediterranean Fever or Behcet disease where the standard dose is 1 mg once daily in adults patients. 

According to table 1, 33 patients had IL-6 dosed at admission but it is not reported the average value.

Response: The IL-6 values are reported in the results section, Lines 267-269. “Unlike patients in the control group, those who received colchicine had IL-6 serially measured because of the physicians’ expectation that the drug would decrease IL-6 levels”, patients in the control group did not have IL-6 values measured. IL-6 levels in the colchicine group are reported in Figure 4. We corrected the typing error in Table 1.

Colchicine may increase the risk of bacterial pneumonia. In this view, even though I admit that the diagnosis is very challenging in this scenario, authors should state if any patient had a suspicion of bacterial pneumonia at baseline during the study follow-up.

Response: This is an important issue. Almost all the patients received antibiotic therapy at the time of COVID-19 diagnosis, but we cannot formally rule out the presence of superimposed bacterial pneumonia. However, in light of the reduced mortality in patients who received colchicine, this hypothesis seems unlikely.

We thank the Reviewer for the constructive comments that helped us improving our paper.

---

## [Decision Letter · Decision Letter 2]

24 Feb 2021

Reduced mortality in COVID-19 patients treated with colchicine: results from a retrospective, observational study

PONE-D-20-29570R2

Dear Dr. Cravedi,

We’re pleased to inform you that your manuscript has been judged scientifically suitable for publication and will be formally accepted for publication once it meets all outstanding technical requirements.

Kind regards,

Antonio Cannatà

Academic Editor

PLOS ONE

Additional Editor Comments (optional):

Reviewers' comments:

Reviewer's Responses to Questions

**Comments to the Author**

1. If the authors have adequately addressed your comments raised in a previous round of review and you feel that this manuscript is now acceptable for publication, you may indicate that here to bypass the “Comments to the Author” section, enter your conflict of interest statement in the “Confidential to Editor” section, and submit your "Accept" recommendation.

Reviewer #2: All comments have been addressed

2. Is the manuscript technically sound, and do the data support the conclusions?

Reviewer #2: Yes

3. Has the statistical analysis been performed appropriately and rigorously? 

Reviewer #2: Yes

4. Have the authors made all data underlying the findings in their manuscript fully available?

Reviewer #2: No

5. Is the manuscript presented in an intelligible fashion and written in standard English?

Reviewer #2: Yes

6. Review Comments to the Author

Reviewer #2: Thank you for addressing all my comments. In my opinion the quality of the paper significantly improved an is now technically sound. The results are of interest and may be useful to better clarify the best therapy of COVID patients

7. PLOS authors have the option to publish the peer review history of their article (what does this mean?). If published, this will include your full peer review and any attached files.

Reviewer #2: No

---

## [Editor Report · Acceptance letter]

15 Mar 2021

PONE-D-20-29570R2 

Reduced mortality in COVID-19 patients treated with colchicine: results from a retrospective, observational study 

Dear Dr. Cravedi:

I'm pleased to inform you that your manuscript has been deemed suitable for publication in PLOS ONE. Congratulations! Your manuscript is now with our production department. 

Kind regards, 

on behalf of

Dr. Antonio Cannatà 

Academic Editor

PLOS ONE